# Two Wheat Cultivars with Contrasting Post-Embryonic Root Biomass Differ in Shoot Re-Growth after Defoliation: Implications for Breeding Grazing Resilient Forages

**DOI:** 10.3390/plants8110470

**Published:** 2019-11-02

**Authors:** Ana Paez-Garcia, Fuqi Liao, Elison B. Blancaflor

**Affiliations:** 1Noble Research Institute LLC, Ardmore, OK 73401, USA; eblancaflor@noble.org; 2Enterprise System and Informatics Department, Noble Research Institute LLC, Ardmore, OK 73401, USA; fliao@noble.org

**Keywords:** roots, wheat, re-growth vigor, grazing, defoliation

## Abstract

The ability of forages to quickly resume aboveground growth after grazing is a trait that enables farmers to better manage their livestock for maximum profitability. Leaf removal impairs root growth. As a consequence of a deficient root system, shoot re-growth is inhibited leading to poor pasture performance. Despite the importance of roots for forage productivity, they have not been considered as breeding targets for improving grazing resilience due in large part to the lack of knowledge on the relationship between roots and aboveground biomass re-growth. Winter wheat (*Triticum aestivum*) is extensively used as forage source in temperate climates worldwide. Here, we investigated the impact of leaf clipping on specific root traits, and how these influence shoot re-growth in two winter wheat cultivars (i.e., Duster and Cheyenne) with contrasting root and shoot biomass. We found that root growth angle and post-embryonic root growth in both cultivars are strongly influenced by defoliation. We discovered that Duster, which had less post-embryonic roots before defoliation, reestablished its root system faster after leaf cutting compared with Cheyenne, which had a more extensive pre-defoliation post-embryonic root system. Rapid resumption of root growth in Duster after leaf clipping was associated with faster aboveground biomass re-growth even after shoot overcutting. Taken together, our results suggest that lower investments in the production of post-embryonic roots presents an important ideotype to consider when breeding for shoot re-growth vigor in dual purpose wheat.

## 1. Introduction

Grazing is a common agricultural practice worldwide that aims to convert plant biomass into animal products such as meat and milk. During grazing, livestock feed on the aboveground part of the plant, which in general leads to the loss of shoot biomass. Loss of aboveground biomass, particularly in situations of heavy grazing pressure, can have adverse effects on pasture stand and livestock productivity. The ability of plants to produce new leaf material after grazing is crucial for their survival, and it is a trait that livestock producers desire because it contributes directly to sustained pasture productivity [1]. To minimize overgrazing, farmers practice rotational grazing, which involves moving animals to a new pasture while the previously-grazed pasture rests. However, rotational grazing can be resource intensive, often requiring large investments in both human and land capital [2]. One potential solution for maintaining pasture productivity and reducing inputs for rotational grazing is to use crop species or varieties that can rapidly re-grow after grazing. 

Crop improvement programs have traditionally been interested in developing cultivars with better re-growth vigor after defoliation by targeting traits that maximize grazing tolerance in perennial and annual grasses. Early studies demonstrated that prostrate and short plant growth result in more leaf area located close to the soil surface. Such traits alleviate the adverse effects of heavy grazing and facilitates rapid re-growth [3]. In addition, maintaining the numbers of buds and tillers is important to increase growth under grazing pressure [1]. Therefore, short genotypes with dense tillering and branching can be successful in cases of intense grazing [1]. However, when controlled grazing practices are used, larger leaves, early-flowering, tall and erect-tillered plants, can be more productive since they provide not only enhanced leaf biomass but also grain yield [4,5]. 

In the Southern Great Plains of the USA and other places around the world, farmers use wheat (*Triticum aestivum*), as a forage and grain crop [6]. Wheat grown in this manner is referred to as dual-purpose and this practice enhances profits compared to a forage-only or grain-only wheat crop [7]. In the USA Southern Great Plains, dual-purpose wheat is planted in early fall (September) and grazed by cattle in late fall and winter before being managed for grain production. Grazing, however, stunts plant development, which in turn delays wheat anthesis and maturity by one day for every 4–5 days of grazing [7,8]. Thus, one characteristic of an ideal dual-purpose wheat cultivar is accelerated re-growth during the late vegetative phase (i.e., during and after the grazing period). Shoot re-growth after grazing not only allows producers to feed livestock for longer periods, but also helps stabilize or enhance grain yield [7,8]. 

A crop’s grazing tolerance is determined by its innate re-growth vigor and by the management practices used by producers. Although plant breeders have been able to develop cultivars with faster re-growth after defoliation, mechanisms underlying such trait remain poorly understood. In efforts to understand mechanisms underlying re-growth after defoliation, physiological changes in grasses exposed to different grazing intensities have been investigated. For instance, reduction of plant photosynthetic capability due to defoliation alters the balance of nitrogen and carbon allocation within plant tissues, and in the plant–soil interface. Destabilizing nitrogen and carbon balance impair the re-growth of the aerial parts making the plant more vulnerable to stress and eventually causing death [9,10]. Other important substrates that determine re-growth vigor after defoliation are the non-structural carbohydrates located in tillers. The amount and nature of non-structural carbohydrates in tillers determine re-growth capability of grasses after grazing [11]. High grazing intensity leads to lower capacity to replenish these carbohydrate reserves during the grazing season. Lower carbohydrate reserves can have a negative effect on forage production and grain yield. 

There is evidence that reduced plant photosynthetic tissue due to grazing has a direct effect on root growth [8]. Reduced root growth could in turn hinder the ability of the plant to resume shoot development leading to poor pasture stand. Studies using leaf clipping as a surrogate for grazing, have found that increased frequency and severity of clipping decreased total root biomass [12,13] and root length [14]. In one specific example, leaf cutting to a height of 100 mm above the soil surface, allowed plants to produce 24 g more root biomass compared to plants cut as low as 12 mm above the soil surface [15]. In another study, Crider [16] used various phenotyping methods to follow root growth over time in a group of grass species subjected to different intensities and rates of forage removal. The author concluded that removing more than 50% of forage in a clipping event causes root growth stoppage for as long as 18 days. In cases of heavy leaf removal, it was found that roots are only able to resume growth one month after cutting. If less than half of the forage is removed, roots grow uninterrupted, which is translated to faster re-growth of the aerial part. Allowing time in between harvesting events, or reducing the amount of aerial parts removed, did not jeopardize forage production or the development of a functional root system [12,13,14,17].

While several studies have demonstrated the effects of defoliation on total root biomass, very little is known about specific changes that occur in the root system when plants are subjected to leaf clipping. In one study, Parker and Sampson [17], found that clipped plants exhibited a reduction in root diameter, radial root area, stele diameter, and the number of vascular vessels. Changes in root anatomy resulted in a poorly developed root system with reduced absorptive and exploratory capacity [17]. Consequently, decreased root system uptake capacity impaired shoot re-growth. 

A common theme among the abovementioned studies is that a tight relationship exists between defoliation and root growth, suggesting that improving shoot re-growth capacity in forages after grazing might benefit from a more in depth understanding of root responses to defoliation. In this paper, we present new insights into how aboveground defoliation affects root development in wheat. Using two wheat cultivars with contrasting shoot and root biomass, we found that leaf clipping reduced post-embryonic root development and modified nodal root angle. Importantly, the cultivar Duster, which had a lower investment in the production of post-embryonic roots, was able to reestablish its root and shoot system more rapidly after defoliation compared to the cultivar Cheyenne, which had a more extensive post-embryonic root system. Our results are consistent with previous studies suggesting that reduced metabolic cost of producing roots (i.e., cheap root ideotype) for soil exploration could benefit shoot re-growth vigor in dual purpose wheat [18].

## 2. Results

### 2.1. Two Wheat Cultivars with Contrasting Forage Productivity in the Field Presented Differences in Root Features and Shoot Biomass in Controlled Greenhouse Conditions

A question that we addressed in this study is whether defoliation in wheat, which is used as a forage crop in the USA Southern Great Plains and other regions around the world, affects root development. We also asked whether the impacts of defoliation on root growth influences shoot re-growth vigor. To answer these questions, we first describe main features of wheat roots focusing specifically on root types (i.e., embryonic and post-embryonic roots) and root system architecture. 

For simplicity, wheat roots have been classified into two main types based on their origin [19,20]. Roots that form during the initial stages of seedling development are called embryonic or seminal roots. Based on this classification, seminal roots emerge from the embryonic tissue as the seed germinates (Figure 1). Embryonic roots play an important role in plant establishment during the first weeks after seed germination because they enable the seedling to explore the soil for water and nutrients before the plant attains full photosynthetic capacity. In wheat, embryonic roots can grow into deeper layers of the soil to maximize resource acquisition [21]. By contrast, post-embryonic roots originate mainly from shoot tissue such as the coleoptile and leaves, or from pre-existing roots (Figure 1). Post-embryonic roots that originate from leaf nodes frequently explore the top layers of the soil. Another type of post-embryonic roots are the lateral roots (i.e., secondary and tertiary roots), which grow from pre-existing root tissue. 

For this study, we selected two hard red winter wheat cultivars that differed in field forage productivity as inferred from historical data gathered during the grazing season [22]. In the USA Southern Plains, the grazing season occurs in late fall until late winter, which typically span the months of November to the end of February [22,23]. The first cultivar used in this study, Duster, is broadly grown in the Southern Great Plains. Released in Oklahoma in 2006, this cultivar has proven to be well adapted to climatologic conditions and management systems characteristic of Southern Great Plains agriculture. Duster’s adaptability has made it a favorite as a dual-purpose wheat crop [24]. The other wheat cultivar selected for this study is Cheyenne. This cultivar was released in Nebraska in 1922 [25]. Although it has been reported that Cheyenne was used as parent of many hard-red winter wheat lines developed in USA breeding programs, its low forage productivity and low resilience to environmental stresses such as drought have limited its use as a forage in the Southern Great Plains [22].

Given the contrasting stress resilience of Duster and Cheyenne, we hypothesized that these cultivars might exhibit differences in their root systems and therefore partly explain why one cultivar (i.e., Duster) is more productive as a forage than the other (i.e., Cheyenne). To test this hypothesis, we quantified seven root traits in Duster and Cheyenne plants grown in the greenhouse. Shoot biomass and tiller number were also quantified to determine if differences in shoot biomass between these two cultivars as reported from field data [22], are also manifested under our controlled greenhouse conditions (Table 1). 

Our results showed that Duster and Cheyenne plants grown in the greenhouse exhibited differences in shoot and root growth. In non-clipped plants, root biomass, root density, and total number of nodal roots in Cheyenne was two to three times higher than Duster (Figure 2A–D). While large differences in previously-mentioned root traits between Duster and Cheyenne were observed, the cultivars exhibited similarities in other root traits. For example, we measured root angle at various points along the first 10 cm of the root system. We found that root angle at different depths was not significantly different between cultivars. Surface root angle of 10-week-old unclipped Duster and Cheyenne plants decreased from 90 to 50 degrees with increased depth, with an average of 70 degrees for the first 3 cm of the root system. Alternatively, root angle in deepest layer (i.e., 10 cm) decreased from 50 to 20 degrees in both cultivars (Figure 3A–C). 

Differences in growth between Duster and Cheyenne in the greenhouse were also manifested in their shoots. In the greenhouse, Cheyenne produced more than twice the shoot biomass, and almost three times the number of tillers than Duster (Figure 4A,B). 

In order to understand the effect that cultivar, treatment, and the combination of both cultivar and treatment had in the shoot and root traits studied, we performed a two-way ANOVA test (Table 2). That statistic method allowed us to have an overview of the combined effect of different leaf clipping intensities and the genetic background of the studied wheat cultivars in the studied shoot and root traits, before we addressed the detailed study of those traits.

Table 2 shows that maximum rooting depth, number of seminal roots and surface root angle were independent of the effect of both genotype and defoliation intensity. In addition, the combination of both factors did not have any effect on these three root traits. Root angle in depth was affected by the defoliation intensity, buy was not determined by cultivar or by the combination of cultivar and treatment. Other root traits that were not affected by the combination of treatment and genotype were number of tillers, number of nodal roots (both total root number and the number of long nodal roots) and root density. These traits, in contrast, were highly defined by the cultivar type, and were also significantly affected by the defoliation treatment. Both shoot and root biomass were determined by the cultivar type, the defoliation treatment and by the effect of both cultivar and treatment (Table 2).

### 2.2. Maximum Rooting Depth and Seminal Root Production Were not Affected by Defoliation

Having established that Duster and Cheyenne exhibited differences in root and above-ground biomass in the greenhouse, we next asked if such differences corresponded to differences in root and shoot re-growth after defoliation. Crider [16], in his study of the effect of leaf clipping in grass root systems, concluded that the amount of leaf biomass removed influenced the extent of root growth stoppage [16]. That study showed that roots of plants with more than 50% of their leaf biomass intact were still able to maintain uninterrupted root growth [16]. Using the Crider (1955) study as a guide, we applied two different leaf removal treatments to Duster and Cheyenne, i.e., one leaf clipping treatment that removed 100% percent of the total shoot biomass and the other treatment that left 60% of the leaves intact. Following Crider’s methodology, we performed three successive clipping events with recovery periods between them [16]. We reasoned that this method would roughly mirror leaf removal patterns during grazing. 

For our study, we focused on the seven root traits noted in Table 1. Among the root traits studied, it was found that maximum rooting depth and seminal root number were not significantly affected by the removal of leaf biomass (Table 2). The maximum rooting depth of Duster was about 100 cm, and this value did not significantly change when leaves were clipped at 40% or 100%. Duster produced close to six seminal roots on average. Like maximum rooting depth, the number of seminal roots in Duster was not affected by leaf clipping. Maximum rooting depth and seminal root number in Cheyenne was not significantly different from Duster. Like in Duster, these root traits were not affected by leaf clipping in Cheyenne (Table 3). 

### 2.3. Cheyenne and Duster Exhibited Differences in Root Biomass, Root Density, Total Nodal Root Number, and Number of Long Nodal Roots in Response to Defoliation 

Although leaf clipping did not affect maximum rooting depth and number of seminal roots, we found that other root architectural traits were modified. One of the most significant effects of leaf clipping was observed on total root biomass. We found that cutting 40% of the leaves caused a 35% reduction of root biomass in Duster, and 55% reduction of root biomass in Cheyenne. When 100% of the leaves were removed, Duster exhibited a 68% reduction of root biomass while Cheyenne had an 83% reduction. Subsequent statistical analyses using one-way ANOVA and Tukey’s tests revealed that clipping-induced root biomass reduction was only significant in Cheyenne, but not in Duster (Figure 2A). 

We next asked, which specific root traits contributed to the observed reduction in root biomass after defoliation. Given that number and length of embryonic roots were not affected by defoliation, we hypothesized that the reduction in total root biomass could be due to impaired post-embryonic root development. To test this hypothesis, we manually counted the total number of nodal roots and the number of nodal roots that elongated more than 38 cm. We found that defoliation significantly decreased the number of both total and long nodal roots in both Duster and Cheyenne (Figure 2B,C). With regard to the total nodal root number, Cheyenne had twice the number of nodal roots than Duster before leaf removal. After cutting 100% of the leaves, total nodal root number in Cheyenne was reduced to similar levels as in Duster (Figure 2B).

To expand our analyses, we looked at root density as a metric to assess the impact of leaf clipping on root system architecture. Root density represents how much of an area is occupied by roots from the same depth in the soil. In using root density as a metric, we included secondary and tertiary lateral roots in our analyses, which are very difficult to measure manually. To measure root density, we first developed a computing algorithm to automatically detect the amount of root material within the image field of view. Our algorithm was based on previous software used to calculate density of filamentous-actin (F-actin) within individual plant cells [26,27]. The process involved converting colored (red–green–blue, RGB) images taken on a photo stand into greyscale (Figure 2E–G). Greyscale images were then processed using the image toolbox of MatLab to eliminate background interference resulting in a final image output in which the root system appeared white within a black background (Figure 2G). The software then recognizes only the white pixels enabling root density to be calculated from the ratio of the area occupied by white pixels over the area of the entire image (see Methods). Consistent with other root traits that we measured manually such as root biomass and total nodal root number, we found that root density of Cheyenne was higher than Duster before leaf clipping (Figure 2D). Defoliation at 40% and 100% led to a reduction in root density in both cultivars. Statistical analysis revealed that reduction was more significant in Cheyenne, compared to Duster (Figure 2D).

### 2.4. Root Angle at Depth Was Affected by Defoliation in Cheyenne, While Root Angle Close to Soil Surface Was not Influenced by Defoliation Treatment or Genotype

Root growth angle is a trait that has been shown to play an important role in crop foraging strategies for water and nutrients because it determines the spatial distribution of the root system within the soil [28,29,30]. We calculated root growth angle by measuring the deviation of the root growth from the vertical. If the displacement from the vertical is high (i.e., wide angle), roots grow along the surface or tend to spread out in a more lateral direction. By contrast, if displacement from the vertical is low (i.e., narrow angles), the root system would grow deeper. 

Because of their thin roots, the 3-D structure of the wheat root system often collapses during excavation from the soil and after washing. As such, obtaining growth angle measurements can be challenging and subject to error. With these caveats in mind, we settled on obtaining 2-D images from the base of the shoot, where the nodal roots grow, and their root growth angle was easy to capture form 2-D images. We focused our measurements on the first 10 cm region of the root system closest to the shoot base. Processed roots were mounted on an imaging box and images were acquired with a digital camera (Figure 3D,E)

We developed a computing algorithm that could automatically detect root growth angle from digitized images of wheat roots. Although there is other published software that enable root angle measurements [31,32], they have some limitations especially for the type of data that we wanted to collect, which involved more spatially resolved root angle distribution throughout the 10 cm root region from the leaf base. Previous software such as DIRT or RootNav focused on acquiring root angles from limited points along the root system [31,32]. With our algorithm, we were able to obtain root angle measurements from the outermost roots that spanned 1 cm intervals of the 10 cm root sample (Figure 3F, see Methods below).

As noted earlier, root growth angle at the surface and in deeper layers of the soil were not significantly different between non-clipped Duster and Cheyenne plants. However, we found that root growth angle was influenced by defoliation. The extent of defoliation-induced root angle modification was dependent on: 1) the amount of shoot biomass removed, 2) the location of the root within the soil profile and 3) the type of cultivar (Table 2). For example, while defoliation at 40% and 100% did not affect root growth angle at the soil surface in both cultivars, it influenced the angle of roots located in deeper soil layers (Figure 3A–C). Root angle at deeper layers (between 6 and 8 cm below the soil surface) became steeper with 100% but not 40% leaf biomass removal. In both cultivars, root growth angle values declined from 35–40 degrees in non-clipped controls to under 30 degrees in plants with 100% of shoot biomass removed. The extent of root growth angle modification after 100% leaf removal was more pronounced in Cheyenne, which exhibited greater than 30% root growth angle reduction compared to the non-treated plants (i.e., 12 degree reduction). By contrast, Duster only exhibited a 21% reduction of its root angle in depth under the 100% clipping treatment (i.e., 8 degrees reduction from non-clipped plants root angle) (Figure 3C). 

### 2.5. Cheyenne Showed Lower Re-Growth Vigor after Shoot Removal Compared to Duster

Having identified root traits that contributed to root biomass reduction after leaf clipping, we next studied shoot re-growth characteristics of 10-week-old Duster and Cheyenne plants following the successive clipping protocol described in Section 2.2 (see also Methods below). 

As noted, non-clipped Duster plants had less shoot biomass compared to Cheyenne under our greenhouse growth conditions. The lower shoot biomass in Duster relative to Cheyenne was also manifested as reduced tiller number (Figure 4A,B). When leaves were clipped at 40% we found that tillers of both cultivars were able to re-grow to similar levels as non-clipped plants. However, only Duster restored shoot biomass to levels comparable to non-clipped plants. When 100% of the leaf biomass was removed, shoot re-growth in both cultivars was extremely low. Although there was partial re-growth one week after the last 100% clipping, shoot biomass levels in both cultivars never reached values of non-clipped plants. At 100% clipping, however, we found that Duster was able to re-grow tillers to numbers close to tiller number of non-clipped plants, suggesting that Duster has a higher re-growth potential than Cheyenne (Figure 4A,B). Because Cheyenne was overall a larger plant than Duster (higher shoot biomass and tiller number) under greenhouse conditions, we expressed biomass and tiller recovery numbers as a percent of starting (i.e., non-clipped) values. In doing so, we found that Duster re-grew almost 50% of its shoot biomass after 40% of the leaves were removed. Recovery was only about 14% when 100% of the leaves were removed (Figure 4C). Percent recovery of Cheyenne shoot biomass was significantly less than that of Duster. Cheyenne only recovered 39% and 8.5% of shoot biomass after 40% and 100% clipping treatments, respectively (Figure 4C). Re-growth advantage of Duster compared to Cheyenne was even more obvious when tiller numbers were expressed as percent recovery. In Duster plants with 40% of their leaves removed, tiller re-growth recovered to about 90% of the original tiller number measured in non-clipped plants. This value was of 44% after 100% leaf-clipping (Figure 4D). By contrast, tiller number in Cheyenne only reached 77% and 38% for 40% leaf-clipping and 100% leaf-clipping, respectively, compared to non-clipped plants (Figure 4D). 

## 3. Discussion

Plant re-growth vigor after grazing is an important trait in forage crops. However, little is known about how grazing influences root system architecture and how grazed-related changes in roots can affect plant development. Therefore, the major goal of this study was to obtain baseline information on the impact of leaf clipping (a surrogate for grazing) on the root system and shoot re-growth of two winter wheat cultivars. Consistent with previous studies, we found that root dry biomass was significantly inhibited by repeated leaf clipping [12,13,15]. However, unlike past studies, which only took into account bulk root traits such as root biomass or total root length [12,13,14,15], we now show the specific root type and corresponding root traits that are modified the most after defoliation. A major finding of our work is that post-embryonic roots are more strongly inhibited by defoliation compared to seminal roots (Table 2). Our results are reminiscent of observations from genetic studies in wheat in which it was shown that a mutation in the tiller inhibition gene (*TIN*) was accompanied by reduced number of postembryonic roots [33].

Our conclusion that seminal roots are less affected by defoliation compared to postembryonic roots is supported by a couple of observations. First, direct counts of seminal root number in both Duster and Cheyenne revealed no differences before and after leaf clipping. Second, a trait that is specified primarily by seminal roots such as maximum rooting depth, was not affected by leaf clipping (Table 3). By contrast, a significant effect of defoliation was observed on nodal root number (i.e., both total nodal root number and number of long nodal roots). Although nodal root number was reduced by leaf clipping, statistical analysis revealed that significant differences were only apparent in Cheyenne. This observation is consistent with the fact that bulk root traits such as root biomass and density, were also significantly reduced in Cheyenne but not in Duster (Figure 2).

An additional root trait that has never been studied with regard to defoliation, is root angle. In recent years, root growth angle has been considered as a target trait in crop breeding due to its role in defining the resource exploratory potential of the root system [28,34]. We found that defoliation caused changes in root growth angle in both Duster and Cheyenne, but only for roots found in deeper (8 to 10 cm from shoot base) layers. Like other root traits, however, statistically significant differences in root growth angle at depth were only observed in Cheyenne clipped plants (Figure 3). At first glance, our results could mean that defoliation limits the lateral root exploratory area by reducing root growth angle and decreasing the total number of post-embryonic roots. An alternative explanation is that the reduction in root growth angle and total number of nodal roots is a strategy to cope with the stress brought about by defoliation. The fewer nodal roots and steep root growth angles observed after defoliation corresponds to the “steep, deep and cheap” root ideotype that has been proposed for plants subjected to abiotic stress [28]. Such an ideotype allows the plant to optimize nitrogen and water acquisition by reaching deeper layers of the soil with minimal investment in root biomass production [18,28,35]. Our results suggest that grazing might activate signaling pathways that overlap with low water and nitrogen stress leading to the “steep and cheap ideotype”. Although we found that root growth angle in depth became steeper after defoliation, there maximum rooting depth was not affected (Table 2 and Table 3). Because plants were grown in mesocosms, it is possible that potential for deeper root growth was suppressed in this system, which prevented us from making more accurate observations on how root growth angle affects rooting depth after defoliation. Stronger associations among grazing, root growth angle and rooting depth will have to be addressed by future studies of wheat plants grown in the field.

Under the greenhouse conditions used in our study, Cheyenne had higher pre-defoliation root and shoot biomass than Duster. Perhaps one reason why statistically significant reductions in post-embryonic root growth after defoliation were only observed in Cheyenne was due to such differences in pre-clipping plant biomass. Because Cheyenne had already invested a significant amount of resources to accumulating biomass prior to clipping, its ability for re-growth could be compromised. This notion is supported by mathematical modeling of plant responses to herbivory. Using such a modeling approach, it was predicted that genotypes with faster growth rate in undisturbed conditions would be less tolerant to increasing percentages of tissue removed by herbivory. Such a prediction was based on the quantification of the fitness reduction over the proportion of plant biomass removed by herbivore damage. Plant fitness is determined by the plant’s ability to perform well in different environments, and depends on the plant intrinsic vigor [36,37]. Our results coincide with this mathematical model in the way that the wheat cultivar with higher root and shoot growth in non-clipping conditions (i.e., Cheyenne), had the largest reduction in shoot and root biomass after leaf removal. By contrast, the cultivar with more conservative growth (i.e., Duster) can resume shoot and root growth after defoliation because of lower investments in total biomass prior to leaf removal. The fact that Duster exhibited minimal and no statistically significant changes in most of the root traits examined after defoliation suggest that this cultivar has less plasticity than Cheyenne. In this regard, it has been proposed that stable growth rate under variable environmental conditions (i.e., lower plasticity) can be an indicator of plant fitness and thus determine plant intrinsic vigor [36,37]. Our results suggest that wheat cultivars with slower root growth under non-stressed conditions could maintain stable root growth when subjected to leaf removal and be more resilient to grazing. The potential advantage in plant vigor under grazing that Duster has over Cheyenne is also apparent in shoot growth recovery after leaf clipping. Duster had lower pre-clipping shoot biomass than Cheyenne. Like its root system, Duster, but not Cheyenne, was able to restore shoot biomass after leaf clipping to almost pre-defoliation levels and could explain why Duster has emerged as a preferred dual-purpose wheat cultivar in USA Southern Plains agriculture [24]. We realize, however, that our observations are based on studies of two cultivars grown in container-based systems in the greenhouse. Nonetheless, our findings lay the groundwork for future studies in field-grown wheat plants and serve as a guide for crop improvement programs of annual cereals crops used for forages. 

In summary, our study has provided information on how certain root traits in dual-purpose wheat respond to leaf clipping. We found that the reduction of root biomass and density that result from leaf clipping in wheat is due to inhibited post-embryonic root growth. In addition, our studies comparing two cultivars with contrasting plant biomass and field forage performance suggest that genotypes with lower number of nodal and lateral roots could be potential ideotypes for grazing-based cropping systems. Genetic modification of the cereal root system has proven to be efficient strategies to improve water and nutrient uptake [34,38,39,40]. However, the relationship between shoots and roots, and how they are affected by stresses associated with grazing, is complex, and likely involves several factors. Knowledge gained from our studies pave the way for more in-depth understanding of shoot and root re-growth after grazing, and how they influence plant vigor in pastures. 

## 4. Materials and Methods 

### 4.1. Experimental Design

Figure 5 shows the experimental layout and workflow of our study.

A mesocosm system [41] composed of 42 Polyvinyl chloride (PVC) pipes of 1.5 m of height and 15 cm diameter was used for this experiment. Pipes were internally covered with transparent plastic bags and filled with Turface and sand (3:1) mixture. 

Pipes were allocated in a greenhouse room with semi-controlled environment. The heating and cooling set points were 23 °C/19 °C day and night. The humidity oscillated from 30% to 60%. The room is equipped with high-intensity discharge lamps of 400 W that were set to come on at 6:30 am and go off 30 min after dusk. Those conditions meant a schedule of 14 h of light/10 h of dark for the first experimental replicate (June to September), and 12 h of light/12 h of dark for the second replicate (September to November).

Pipes were divided in two groups of 21. Each group was planted with a different wheat genotype. Each block of 21 pipes was randomly located in the greenhouse and appropriately labelled (Appendix A). Seven pipes were assigned to each one of the different treatments, i.e., “No Clipping” (NC), “40% of leaves clipped” (40%C) and “100% of leaves clipped” (100%C). Each pipe was watered until water fully drained out from the bottom of the pipe to assure homogenous moisture along the soil column. 

Fifty wheat seeds per cultivar were pre-germinated for three days in between two wet germination paper sheets. Once pre-germinated, forty-two seedlings of the same size were selected for planting. Two seedlings were planted per pipe, and then thinned to one at the two-leaf stage after seven days [42] to assure that only one plant per pipe was studied. Duster and Cheyenne blocks were planted with two days of difference to allow enough time for harvesting and measuring processes between blocks.

The experiment was repeated twice to eliminate the effect of the greenhouse temperature and light variations in plants growth. Thereby, a total of fourteen samples per each cultivar and clipping treatment, from two independent experiments were used in this study. In the first replicate, two plants died, and two more plants were lost in the second replicate. That left a total of thirteen samples for Duster-No Clipping, Cheyenne—40% Clipping, Cheyenne—100% Clipping and Duster—100% Clipping. Cheyenne-No Clipping and Duster—40% Clipping were composed of 14 total samples.

### 4.2. Shoot and Root Phenotyping Methodology

Plants were allowed to grow for five weeks before clipping. Plants were watered from the top surface every two days. Seven plants of each genotype for each replicate were subjected to the clipping treatments: no clipping, 40% and 100% clipping of leaves and stems. Leaf clipping through the whole experiment was done starting from the oldest tiller (the one closer to the soil surface) and continuing to the newer ones. In each tiller, we started cutting leaf blades above the oldest leaf node, and continued with the newest ones. In the case of plants in which 100% of the leaves were removed, all the green material (leaves and stems) above the leaf node closer to the soil surface was cut.

Seven weeks after planting, the second cutting was performed following the same methodology described above. At this stage, pipes were individually watered using 900 mL of water per pipe every two days. Once a week plants water was complemented with 15–30–15 Water Soluble Fertilizer (Scotts), in a concentration of 0.2 g per liter of water. After nine weeks from planting, the third cutting was performed as described above. Plants were allowed to grow for one more week after the third cutting, for a total of 10 weeks. Plants were then harvested.

Each plant was individually extracted from the pipe and placed on top of a 2 mm mesh for washing (Appendix A). This washing system prevented loss and breakage of thinner roots, and allowed us to recover the whole root system in one piece. This practice is very important in order to have precise measurements of root traits as maximum rooting depth, for instance.

The entire plant was photographed (Appendix A). Number of tillers was measured and shoots were separated from the roots. Shoots were placed in paper bags and dried at 37 degrees for three weeks in a Constant Temperature Oven DKN192 (Yamato Scientific America Inc, Santa Clara, CA, USA). After three weeks, the dry material was weighted. Roots were detailed washed to remove all the remaining debris (Appendix A). Primary seminal (embryonic) and primary nodal (post-embryonic) roots were quantified. Primary roots are the ones that directly develop from embryo or leaf nodal tissue. These roots are thicker and easy to recognize for the trained eye.

### 4.3. Computing Software for Root Density and Root Angle Analysis

After washing and manually counting number of primary seminal and primary nodal roots, the first 20 cm of the root system was cut and kept for further analysis, while the remaining root system was bagged and dried. 

Roots were spread in a transparent plastic tray filled with water. The tray was placed on top of a trans-illuminator, so the light hit the bottom of the tray. This imaging system allowed to get contrasted images of the root system using a camera located on a vertical stand (Figure 2E,F). All the root samples were cut at the same maximum length (20 cm) and the total area imaged was the same for all the samples (550 cm^2^, area of the plastic tray), so the main differences between images where due to the number of primary, secondary and tertiary roots. A new computing software was developed to analyze root images. The software was written in MATLAB 2018, using image-processing toolbox. Initially, the program converted RGB images into gray style and divided the gray style into 96 layers based on intensity distribution, which ranged from 0 to 255. A threshold was applied to the multi-layer images, in order to cancel out the background interferences. The output of this process were images where the root system is highlighted in white color (Figure 2G). The software then measured the number of white pixels as root area, and the root density was calculated as ratio of root area by area of the image. Using this system, the area occupied by roots (primary, secondary and tertiary) was measured in each image. The software allowed a consistent high-throughput quantification of root system branching density, so average root density can be compared between genotypes and treatments.

After the roots were processed for root density measurements, the first 10 cm of the root from the plant crown was cut and kept for root angle analysis. The rest of the root system was bagged and dried. Roots hanging inside a portable light box were imaged to obtain root images with black background, reducing background interference (Figure 3D,E). Using MATLAB 2018 with image-processing toolbox, a new-developed software measured two curves starting from the vertex, following the most externally situated roots on the right and left side of the root system, and stopping at the root tip (Figure 3F). Root angle was defined as the angle between two intersecting lines that started from the vertex and extended separately to the left and right curves, ending at the same vertical distance from vertex. The software calculated root angle in degrees continually from the position right below the vertex (vertical distance closer to zero) to the root tip (Figure 3F).

### 4.4. Statistical Analysis

For all treatments, the number of individual samples analyzed were N ≥ 13 per treatment, from two independent experiments. For data collection and averages, standard deviations, standard errors of the mean and t-test calculation were used Microsoft Excel 2013 and 2016 for Windows.

The effect of the different treatments in the studied cultivars was compared by contrasting averages using one-way ANOVA test. Then, the cultivars submitted to different treatments were categorized in groups regarding the statistically significant differences in their averages, using Tukey’s test. For one-way ANOVA test and Tukey’s test we used IBM SPSS software platform.

KaleidaGraph software was used to plot data into bar graphs.

## 5. Conclusions

Post-embryonic roots (i.e., nodal and lateral roots) were particularly affected by leaf clipping in two wheat cultivars. This root type can be of special interest for breeding programs oriented to maximize wheat productivity in grazed-based farming systems. 

Leaf removal causes a reduction in total root system exploratory area by reducing number of roots. In addition, plants subjected to defoliation presented steeper root system.

Wheat cultivars with reduced root growth (mainly nodal and lateral roots) are more likely to present increased resilience to grazing.

## Figures and Tables

**Figure 1 plants-08-00470-f001:**
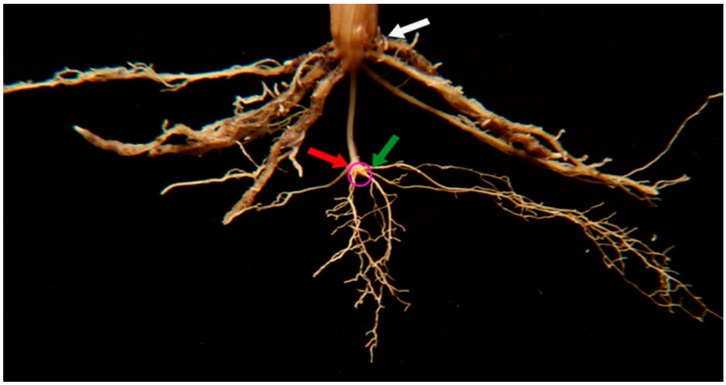
Root system of a two-month-old wheat plant excavated from the field. The location of the embryo after seed germination is marked with a magenta circle. The green arrow indicates the location of a seminal root, the root type that emerges from embryonic tissue during germination. The red arrow points to a post-embryonic root, which originates from coleoptile tissue. The white arrow marks a post-embryonic root that originates from the base of the leaf node.

**Figure 2 plants-08-00470-f002:**
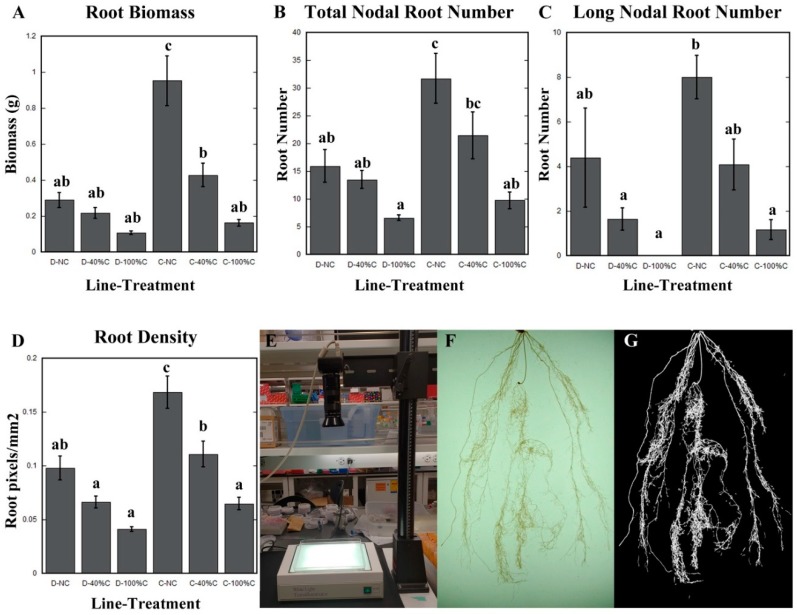
Effect of defoliation on selected root traits of two winter wheat cultivars. (**A**) Root biomass. (**B**) Total nodal root number. (**C**) Long (>38 cm) nodal root number. (**D**) Root density calculation based on occupancy analysis. (**E**) Roots were spread in a plastic tray illuminated from above to obtain photos with high contrast. The camera was mounted on a vertical stand to acquire images of the whole root system. (**F**) Representative pictures of the first 20 cm of the root of a 10-week-old wheat plant, obtained with the system shown in (**E**). (**G**) Corresponding segmented image of the roots in (**F**) for measuring root density. No leaves clipped = NC, 40% of the leaves clipped = 40% C, and 100% of the leaves clipped = 100% C. Duster = D and Cheyenne = C. Error bars represent standard error of the mean. N ≥ 13 ten-week-old plants. Statistically significant differences of the means were determined using One-way ANOVA and Tukey’s tests. Different letters represent distinct means within groups at *p* < 0.05.

**Figure 3 plants-08-00470-f003:**
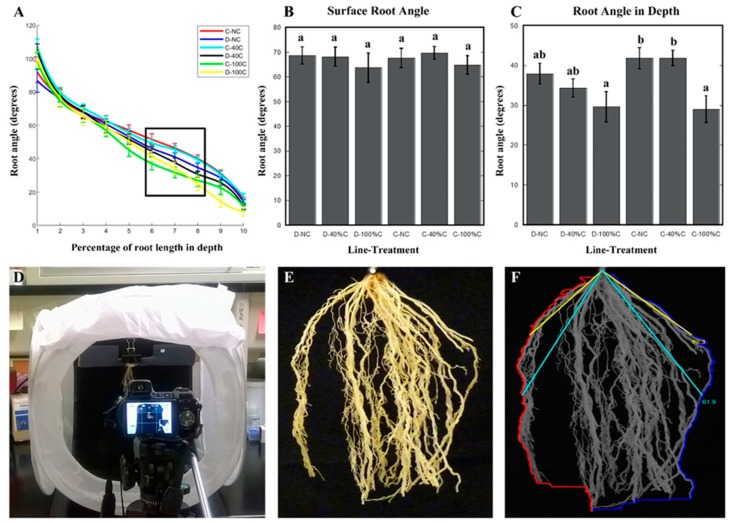
Root angle characterization in two wheat cultivars subjected to defoliation. (**A**) Root angle distribution in depth. Roots where divided into 10 segments of 1 cm each. The graph shows the average root angle within these segments. The different colored lines represent cultivar-treatment combinations. The black square marks the root area where angle values differ the most, i.e., between the sixth and the eighth segments. (**B**) Surface root angle distribution. The graph represents the root angle average of the first three segments of roots. (**C**) Root angle distribution in depth. Root angle average between 6 and 8 cm segments was calculated and represented per each cultivar and treatment. (**D**) Image acquisition system for root angle measurements. (**E**) Representative photo of the first 10 cm of the root of a 10-week-old wheat plant, obtained with the system shown in (**D**). (**F**) Software used to quantify root angle in wheat roots. The crown of the plant was chosen as the vertex. From the vertex, two lines (blue and red in **F**) were drawn to mark most external roots. Root angle was defined as the angle in degrees between two intersecting lines that started from the vertex and extended separately to the left and right curves at the same vertical distance from vertex. Yellow and cyan lines in (**F**) represent two examples of root angle measurements. Yellow lines show root angle at 25% of vertical root length. Cyan lines represent the angle at 50% of vertical root length. No leaves clipped = NC, 40% of the leaves clipped = 40% C or 40 C, and 100% of the leaves clipped = 100% C or 100 C. Duster = D and Cheyenne = C. Error bars represent standard deviation of the mean. N ≥ 13 ten-week-old plants. Statistically significant differences of the means were determined by using one-way ANOVA and Tukey’s tests. Different letters represent distinct means within groups at *p* < 0.05.

**Figure 4 plants-08-00470-f004:**
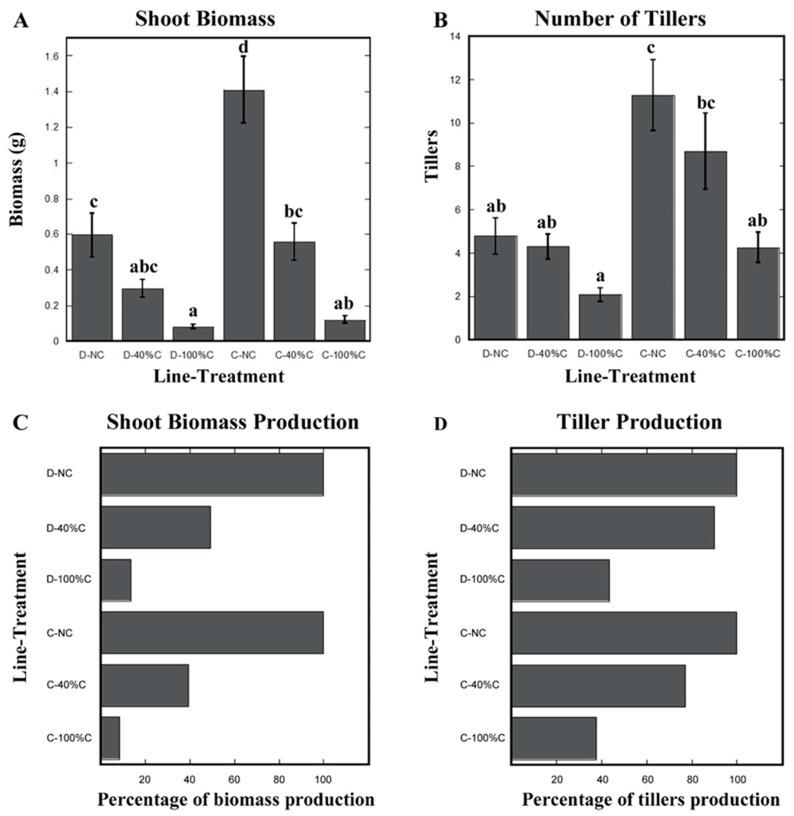
Aboveground biomass re-growth in two wheat cultivars subjected to defoliation. (**A**) Shoot biomass. (**B**) Number of tillers. (**C**) Percentage of shoot biomass recovery in Duster and Cheyenne after leaf clipping, compared to non-clipped plants (represented as 100% of shoot biomass production). (**D**) Percentage of tiller recovery in Duster and Cheyenne after leaf clipping, compared to non-clipped plants (represented as 100% of tiller production). No leaves clipped = NC, 40% of the leaves clipped = 40% C, and 100% of the leaves clipped = 100% C. Duster = D and Cheyenne = C. Error bars represent standard deviation of the mean. N ≥ 13 ten-week-old plants. Statistically significant differences of the means were determined using one-way ANOVA and Tukey’s tests. Different letters represent distinct means within groups at *p* < 0.05.

**Figure 5 plants-08-00470-f005:**
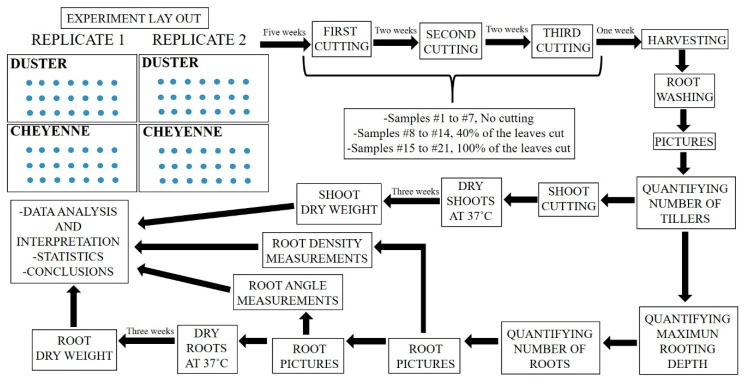
Experimental design flowchart. Arranged clockwise, the chart represents the experimental steps followed in this study. On the left top corner is represented the random distribution of Duster and Cheyenne treatments in two independent experiments (replicates 1 and 2). Reading from left to right, we show the experimental procedure including leaf cutting, plant harvesting and root and shoot traits quantification. On the bottom left corner, the final steps of the study are illustrated, such as data analysis and interpretation, and conclusions extraction.

**Table 1 plants-08-00470-t001:** Shoot and root traits quantified in Duster and Cheyenne. The first column lists the specific shoot and root traits quantified in the experiment. The second column shows corresponding units for each trait, when applicable. See Section 4 for a detailed explanation on methods for trait measurement.

TRAIT	UNITS
Number of tillers	n/a
Shoot biomass	Grams (g)
Root biomass	Grams (g)
Maximum rooting depth	Centimeters (cm)
Number of nodal roots	n/a
Number of seminal roots	n/a
Surface Root Angle	Degrees
Root Angle in depth	Degrees
Root Density	Pixels/millimeter square (px/mm^2^)

**Table 2 plants-08-00470-t002:** Effect of cultivar, treatment and the combination of both factors in the studied shoot and root traits. The first column lists the specific shoot and root traits quantified in the experiment. Second to fourth columns show the effect of cultivar type (i.e., Duster or Cheyenne), treatment (i.e., no leaf clipping, 40% leaf clipping or 100% leaf clipping) and the combination of both in the quantified traits showed in the first column. Significant differences of the means were determined by using two-way ANOVA. Asterisks represent distinct means within groups at *p* < 0.05 (*); *p* < 0.01 (**); *p* < 0.001 (***) and no significant effect (ns).

Trait	Cultivar	Treatment	Cultivar * Treatment
Number of tillers	***	***	ns
Shoot biomass	***	***	**
Root biomass	***	***	***
Maximum rooting depth	ns	ns	ns
Number of nodal roots	***	***	ns
Number of long nodal roots	*	***	ns
Number of seminal roots	ns	ns	ns
Surface Root Angle	ns	ns	ns
Root Angle in depth	ns	**	ns
Root Density	***	***	ns

**Table 3 plants-08-00470-t003:** Maximum rooting depth and number of seminal roots were not affected by defoliation. The first two rows of the table specify root traits and statistics of the mentioned traits (average ± standard error of the mean (SEM)). The first column on the left represents different treatments applied. There were not statistically significant differences among the treatments (*p* < 0.05).

ROOT TRAIT	Maximum Rooting Depth (cm)	Number of Seminal Roots
	Average ± SEM	Average ± SEM
Duster NO clipping	100.1 ± 5.2	5.7 ± 0.3
Duster 40% clipping	106.7 ± 5.8	5.8 ± 0.3
Duster 100% clipping	105.4 ± 8.0	5.9 ± 0.3
Cheyenne NO clipping	98.2 ± 3.8	5.99 ± 0.2
Cheyenne 40% clipping	94.2 ± 4.0	5.4 ± 0.2
Cheyenne 100% clipping	96.8 ± 4.3	5.7 ± 0.3

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
