# Peer review of "Two Wheat Cultivars with Contrasting Post-Embryonic Root Biomass Differ in Shoot Re-Growth after Defoliation: Implications for Breeding Grazing Resilient Forages"

_plants, 2019, doi:10.3390/plants8110470_

Round 1

Reviewer 1 Report

This manuscript presents an interesting study of the impact of grazing on root development and growth recovery of two wheat cultivars. The subject is well presented and the experimental approach is relevant, with many root parameters taken into account. I will only have two main remarks: I think that the structure of the ‘results’ section should be slightly revised, some paragraphs should be put in the ‘introduction’ section (where currently this information is missing, like the description of the two cultivars or the initial hypotheses) while others would be more in place in the ‘Mat&Meth’ section (for ease of readability of the result section); see comments below. My second concern is about the type of statistical analysis performed: I think that more justifications should be given to explain the choice to compare all ‘cultivar x treatment’ combinations together. Why not have made a two-way anova with cultivars and treatments separately? This could have sometimes simplified the results obtained by determining whether it was only the effect of the treatment (or cultivar) or whether the cultivars had a different response to a same treatment. If there was an interest in really observing the evolution of each cultivar compared to applied treatments, why not have analyzed the effect of treatments on Duster and Cheyenne separately? Some results (Figure 2, Figure 4) would probably have been statistically different.

I really appreciated the many illustrations of the results and the methods, in particular the graphic abstract. For this one, I suggest some modifications to make it even easier to understand (without having to read the legend): Intuitively, I would think that 'shoot biomass' or 'root density' etc. are better capacities whereas it is the opposite. It might be interesting to clearly identify 'no cutting' as the reference level (and that comparisons are with this control and not one cultivar compared to the other). So, perhaps add the sign "-" in front of the traits or add this clarification in the title, something like "Root and shoot re-growth after leaf cutting in two wheat cutivars, i.e. Duster (left) and Cheyenne (right): failing traits compared to no cutting control ".

Some suggestions of paragraphs that can be moved to facilitate the readability of the manuscript:

L109-115: can be removed since it’s only short summary of results that will be developed in the next section. L119-159: would have completely its place in the introduction. It’s a good description of the two used cultivars and the a priori hypotheses. Table 1: can be moved to the ‘Mat&Meth’ section L182-185: could be included in the introduction L185-190 and L238-247 and L265-275: could be included in the ‘Mat&Meth’ section Fig 2 E to G and Fig 3 D to F: could be included in the ‘Mat&Meth’ section

All these suggested changes can lighten the result section and better highlight the new results obtained.  

Minors comments:

Very good idea to make ''highlights results'' titles for each part of the results section. Consider doing the same for 2.1 L124: maybe mentioned the name of the two type of roots at the end of the first sentence, before the description of each. Specify clearly which are the nodal roots which will be indicated later in the text (primary, secondary, tertiary post-embryonic roots?) Consider adding the age of the seedlings in the caption of the figures Table 2: line 2, SEM of Number of seminal roots, add a ± L230: why a threshold at 38 cm? L361: Table 2 instead of Table 1 L466: 7 plants of each genotype for each replicate?

Author Response

The authors appreciate the thorough revision of Reviewer 1. The proposed changes will improve the manuscript quality and readability.

We found that the suggestion of moving part of the first section of the results to the Introduction can make the Introduction very heavy. As Figure 1 is an original photo from plants studied in lab, we think that it could facilitate the reader to understand better the type of root traits and root types that will be quantified in the study. For the same reason (to make the rational of the study easy to follow), we think that is better to keep Table 1 at the beginning of the Results sections, so the reader has it present during the results section reading. In addition, we placed the explanation of the software used for root density and root angle measurements in the main results text instead of in the Mat and Meth sections because we wanted to highlight the important of the development of those two new software for the study. We are aware of the existence of other software that can help us measure these root traits, but we decided to create our own pipeline to assure that the measurements in our wheat plants grown in the turface:sand mix where accurate and fit our needs.

We thank the reviewer for the comment on the statistical analysis. We performed a two-way ANOVA that is now represented in Table 2. We did not find additional significant differences using this method, but we think is very visual as summary of the effect that genotype, treatment, and the combination of phenotype and treatment have in the studied traits. We included the corresponding explanation of the table in the Results and in the Discussion sections.

We thank the reviewer kind comment about the addition of illustrations supporting the text. We really think that can help the reader understand different concepts of the study. We address the changes proposed by the reviewer in lines #35 and #36.

We very appreciate the reviewer’s minor comments. We addressed them where appropriate following the next list:

Very good idea to make ''highlights results'' titles for each part of the results section. Consider doing the same for 2.1. We addressed this comment in lines #119-120

 L124: maybe mentioned the name of the two type of roots at the end of the first sentence, before the description of each. Specify clearly which are the nodal roots which will be indicated later in the text (primary, secondary, tertiary post-embryonic roots?)

Addressed in Lines #126 and #134

Consider adding the age of the seedlings in the caption of the figures

Addressed in Figure legends.

Table 2: line 2, SEM of Number of seminal roots, add a ±

Done.

L230: why a threshold at 38 cm?

We cut the plants using an US tape, so we cut the first 15 inch of the root system form the base of the leaves node. When we converted the inches to cm to have all our measurements in the IS, the conversion gave us the odd threshold of 38cm.

L361: Table 2 instead of Table 1

Corrected in line #387

L466: 7 plants of each genotype for each replicate? 

Fixed in line #475

Reviewer 2 Report

Review of the manuscript entitled: ‘Two wheat cultivars with contrasting post- embryonic root biomass differ in shoot re- growth after defoliation: implications for breeding grazing resilient forages’ by Paez-Garcia et al., submitted to Plants.  

The Authors present data and conclude that seminal roots are less affected by defoliation than postembryonic roots of wheat. From the results presented it is clear that a poor root system can cause inhibition of shoot re-growth leading to lower productivity. This work is well documented, interesting and original study that merits publication. Authors used appropriate methods, performed control experiments, and that make the results reliable by statistical analyses applied.

Generally, this work presents well understood concept and well prepared experiment. However, small changes can be made.  I wish Authors do some extra work to strengthen the publication.

Specific remarks:

Introduction is adequate and presents the relevant state of the art. I like the methodology and result presentation form included. Everything is clear and such flowchart as presented on Fig. 5 is really helpful for the potential readers. The discussion is actually quite enjoyable to read, however it seems that it could be more focused on the verification of hypothesis and findings of this actual publication in relation to specific literature in the field. It seems that citation of only 15 papers in discussion is insufficient and inadequate.

References - the list should be corrected according to instruction for authors. Please note that in the list journal name should be abbreviated, title of the publication should not be written in bold, add dots after Authors’ initials etc. Please, do not trust your reference package/manager - you need to edit references to fix the formatting errors.

Author Response

We really appreciate the comments of Reviewer 2, they will definetly will help improving the manuscript.

We would like to explain a few points regarding the reviewer's concern about commenting only 15 papers in the discussion section. To the best of our knowledge, this is the first study adressing the correlation between specific root traits and crop re-growth vigor after defoliation. Previous studies had showed how leaf clipping affects bulky root traits such as root biomass and total root length, and we discuss those studies in the Introduction (lines #87 to #166). We also quantified those root traits in our study, and we properly discussed the similarities with previous work in the Discussion (e.g. lines #352 to #357). However, the main interest of this study has been to explore the correlation between defoliation-induced changes in specific root traits such as number of seminal and crown roots, or root growth angle, and the re-growth vigor of wheat plants after different leaf clipping treatments. Those results are discussed in lines #369 and beyond. However, the lack of previous published results in this regard, makes it dificult to discuss them in such a context. Nevertheless, this novelty is what makes the present manuscript so interesting.

We thank the rewiever for the comment regarding the bibliography. We addressed the mistakes in the format of the list of references. We think that now they fit better the journal requirements.

Reviewer 3 Report

A very well written manuscript, acceptable to the Plants Journal after minor revisions and clarifications.

Edits/corrections/suggestions for the author(s)’ consideration have been added to the attached Manuscript using Track Changes.

Thank you.

Author Response

The authors thank Reviewer 3 for her/his positive feedback and useful suggestions to improve the manuscript.

Following the attached document "peer-review-5365930.v1.pdf", we have incorporated corrections and suggestions in the next lines of the revised manuscript:

38, 44, 92, 95, 102, 183, 334, 348-349, 417, 434, 436-437, 449-451, 454, 456, 457, 469-477, 479-480, 487, 516, 523-524, 535-536 and 549.

We also addressed comments in Table 2 heading and Table 2 layout, legends of figures 2, 3, 4, 5 and S1and content of Figure 5.

We appreciate all of the reviewer's comments and we would like to give a more complete answer to some of them here:

 -Comment AA9, lines 447-449: We cannot be 100% sure that 2 days difference won't affect the results, but the conditions in the greenhouse room we use for this study were quite stable, so we can assure that light, temperature and humidity were very similar for both Duster and Cheyenne cultivars in each replicate. In addition, having that time gap was necessary to assure that roots were adequately processed. We assumed that plant samples were more likely to be affected once that they were extracted from the growing soil, so we wanted to finish that process as soon as possible and have the roots properly storaged for further characterization. 

-Comment AA14, line 480: We apologize for the confusing picture, we modified the figure to make it clearer.

-Comment AA16, lines 517-526: Thanks for sharing your concern with us. We think that we discussed properly the limitations of root angle measurements using a root system like wheat in the section 2.4 of the Results. In order to allow potential readers to benefit from the software created for this study, we will make the code available upon request to the corresponding author, or we will upload it to a github link, whatever the Editors feel fits better with the journal requisitions.

-Comment AA17, line 536: the paragraph has been rephrased for clarity.